# CP12 Is Involved in Protection against High Light Intensity by Suppressing the ROS Generation in *Synechococcus elongatus* PCC7942

**DOI:** 10.3390/plants10071275

**Published:** 2021-06-23

**Authors:** Masahiro Tamoi, Shigeru Shigeoka

**Affiliations:** 1Department of Advanced Bioscience, Faculty of Agriculture, Kindai University, 3327-204 Nakamachi, Nara 631-8505, Japan; 2Agricultural Technology and Innovation Research Institute, Kindai University, 3327-204 Nakamachi, Nara 631-8505, Japan; 3The Experimental Farm, Kindai University, 2355-2 Yuasa, Yuasa-cho, Arida-gun, Wakayama 643-0004, Japan; shigeoka@nara.kindai.ac.jp

**Keywords:** CP12, cyanobacteria, photo-oxidative stress, photosynthesis

## Abstract

We previously reported that CP12 formed a complex with GAPDH and PRK and regulated the activities of these enzymes and the Calvin–Benson cycle under dark conditions as the principal regulatory system in cyanobacteria. More interestingly, we found that the cyanobacterial CP12 gene-disrupted strain was more sensitive to photo-oxidative stresses such as under high light conditions and paraquat treatment. When a mutant strain that grew normally under low light was subjected to high light conditions, decreases in chlorophyll and photosynthetic activity were observed. Furthermore, a large amount of ROS was accumulated in the cells of the CP12 gene-disrupted strain. These data suggest that CP12 also functions under light conditions and may be involved in protection against oxidative stress by controlling the flow of electrons from Photosystem I to NADPH.

## 1. Introduction

Oxygenic photosynthesis by plants, algae, and cyanobacteria converts sunlight into chemical energy, essential for their metabolism. The Calvin–Benson cycle is the primary pathway for reducing CO_2_ into biomass; therefore, the control of this cycle in response to environmental changes or the developmental stage is indispensable for photosynthetic organisms. In the Calvin–Benson cycle, the activities of thiol-modulated enzymes, such as NADP^+^-glyceraldehyde-3-phosphate dehydrogenase (GAPDH), fructose-1,6-bisphosphatase (FBPase), sedoheptulose-1,7-bisphosphatase (SBPase), and phosphoribulokinase (PRK), are activated by the reduction of intramolecular disulfide bridges via the ferredoxin/thioredoxin (Fd/Trx) system during the day [1,2]. When these cysteine residues of the four enzymes are oxidized to form a disulfide bond, they change to the inactive form during the night. The Fd/Trx system is one of the best characterized regulatory mechanisms in plants.

On the other hand, it has been clarified that the enzymes constituting the cyanobacterial Calvin–Benson cycle are not regulated by the Fd/Trx system, unlike those derived from higher plants [3]. These cyanobacterial enzymes lack the cysteine residues required for activity control or the different structures near the cysteine residues [4,5,6,7,8]. Instead, the small protein CP12 binds to GAPDH and PRK and regulates the activities of these enzymes in response to changes in intercellular NAD(P)H/NAD(P) levels under light and dark conditions. It was reported that carbon metabolism could be regulated through controlling the activity of these enzymes by associating/dissociating a complex in cyanobacteria [9]. In fact, the cyanobacterial CP12 gene-disrupted strain (Sc∆CP12) grew normally under constant light conditions, but GAPDH and PRK activities could not be controlled under light and dark conditions and showed growth retardation because of abnormal carbon metabolism [9]. From these results, it was clarified that CP12 plays an important role in controlling the carbon flow from the Calvin–Benson cycle to the oxidative pentose phosphate cycle. Moreover, by structural-basis analysis, it has been reported that the CP12, GAPDH, and PRK could integrate each other depend on both redox state and NAD(P)H/NAD(P) levels to regulate carbon metabolism in cyanobacteria [10].

Interestingly, the growth of Sc∆CP12 was clearly slower than wild-type growth under continuous high light conditions. This phenotype cannot be explained by the CP12 function described above; thus, new CP12 functions are expected. In higher plants and algae, multiple CP12 homologs are present and have unique functions other than controlling the activity of PRK and GAPDH [11,12,13,14,15,16,17,18]. It has been reported that CP12-1 and CP12-2 work cooperatively to regulate photosynthetic ability through controlling protein levels of PRK in mature leaves of *Arabidopsis* [19]. However, there have been few reports on the association between CP12 and environmental stress in plants. In the present study, in order to clarify the novel function of CP12 in cyanobacteria, we analyzed the response to high light in Sc∆CP12 mutants.

## 2. Results

### 2.1. The Lack of CP12 Suppressed Growth under High Light Conditions

The growth rate of Sc∆CP12 mutants was compared with that of wild-type cells under photoautotrophic conditions. It has already reported that Sc∆CP12 grows like the WT at 40 μmol photons m^−2^ s^−1^ in Tamoi et al. 2005 [9]. We reconfirmed that the growth rate of the Sc∆CP12 mutant was almost the same as that of wild-type cells under continuous low light conditions at 40 µmol photons m^−2^ s^−1^ (Figure 1A). Under continuous low light conditions at 40 µmol photons m^−2^ s^−1^, the growth rate of the Sc∆CP12 mutant was almost the same as that of wild-type cells (Figure 1A). Next, we compared the growth rate of both types of cells under continuous high light conditions at 100 µmol photons m^−2^ s^−1^. Under these conditions, wild-type cells grew slightly better than when grow at low light conditions. Interestingly, Sc∆CP12 mutant cells grew significantly slower than wild-type cells (Figure 1B).

### 2.2. Effect of High Light Treatment on Chlorophyll Contents

Chlorophyll contents were compared when cells cultured under low light conditions (40 µmol photons m^−2^ s^−1^) until the logarithmic growth phase were transferred to high light conditions (500 µmol photons m^−2^ s^−1^). After 6 h of high light treatment, the amount of chlorophyll in the wild strain was reduced to 84% of that before the treatment. On the other hands, in Sc∆CP12 cells, the amount of chlorophyll decreased to 45% and 27% after the high light treatment for 3 and 6 h, respectively, of that before the treatment (Figure 2).

### 2.3. Effect of High Light Treatment on Photosynthetic Activity

The rates of NaHCO_3_-dependent O_2_ evolution in wild-type and Sc∆CP12 cells under low light (40 µmol photons m^−2^ s^−1^) illumination were 86.3 ± 1.15 and 85.8 ± 1.33 µmol h^−1^ (mg chlorophyll)^−1^, respectively; that is, the former was almost the same as the latter. After high light (200 µmol photons m^−2^ s^−1^) treatment, the rates of NaHCO_3_-dependent O_2_ evolution of wild-type cells decreased to 86% and 80% at 4 and 5 h after the treatment, respectively, while that of Sc∆CP12 cells decreased to 60% and 40% at the same duration (Figure 3).

### 2.4. Effect of High Light Treatment on Intercellular ROS Level

To determine whether the inhibition of growth in Sc∆CP12 cells was due to photo-oxidative stress, the level of intercellular ROS was measured by a ROS-specific reagent (Figure 4). There were no marked differences in ROS levels between wild-type and Sc∆CP12 cells before high light treatment. After high light (200 µmol photons m^−2^ s^−1^) treatment, ROS levels of wild-type cells increased 2.4-, 3.8-, and 5.6-fold at 1, 4, and 8 h after stress treatment, respectively, while that of Sc∆CP12 cells increased 3.0-, 5.9-, and 9.8-fold with the same duration. At 3 h after high light treatment, the ROS level in Sc∆CP12 cells was 3.1-fold higher than that in wild-type cells.

### 2.5. Effect of Paraquat Treatment on Growth of Wild-Type and Sc∆CP12 Mutant Cells

Paraquat (methyl viologen) is widely used as an electron acceptor of Photosystem I (PSI) and inducer of ROS generation. To assay the effect of paraquat on growth, wild-type and Sc∆CP12 cells were cultured in various concentrations of paraquat and exposed to a low light intensity (40 μmol m^−2^ s^−1^) for seven days. The Sc∆CP12 cells showed slight chlorosis with 0.1 µM paraquat compared with wild-type cells. Growth of the wild strain was inhibited as the concentration of paraquat increased, but more marked growth inhibition was observed in the mutant strain, and the mutant strain could not grow in medium containing more than 0.2 µM paraquat (Figure 5).

## 3. Discussion

We previously demonstrated that CP12 formed a complex with GAPDH and PRK and regulated the activities of these enzymes and the Calvin–Benson cycle under dark conditions in cyanobacteria. Moreover, photosynthetic activity of Sc∆CP12 was almost same with that of wild-type under 40 μmol m^−2^ s^−1^, suggesting that the lack of CP12 did not affect the capacity of the Calvin–Benson cycle under light condition [9]. Therefore, CP12 has been considered to work only under dark conditions. Under very weak light conditions, there was no significant difference in the growth of Sc∆CP12 and wild-type cells. When the light intensity was increased, the growth of the wild-type was promoted more than that under the low light condition, but the growth of Sc∆CP12 was clearly lower than that of the wild-type. These data indicate that CP12 must also function under light conditions. As previously reported, the lack of CP12 did not affect photosynthetic activity in cyanobacteria [9]. These results indicate that the growth of Sc∆CP12 was inhibited without a decrease in photosynthetic activity, suggesting that CP12 plays an important role in the adaptation of photosynthesis to increasing light intensity. In photosynthetic organisms, it is well-known that reactive oxygen species (ROS) accumulate depending on the increase in light intensity, causing oxidative damage [20,21,22]. In *Arabidopsis*, it has been reported that CP12 binds to GAPDH and PRK and protects SH groups involved in regulating the activity of these enzymes from the oxidative state under oxidative stress conditions [23]. However, we previously demonstrated that the activities of GAPDH and PRK were not regulated by the Fd/Trx system via these SH groups in cyanobacteria [4,7,8]. Therefore, increased stress sensitivity is not considered to be due to a lack of SH group-protection ability by CP12, suggesting the possibility of a novel defense mechanism involving CP12 in cyanobacteria. Moreover, there are no reports that CP12 deficiency increases susceptibility to oxidative stress in higher plants.

In order to clarify the function of CP12 to protect against high light conditions, various parameters were compared when cells cultured under low light until the logarithmic growth phase were irradiated with high light. As a result, irradiation with 500 µmol photons m^−2^ s^−1^ led to a slight decrease in the amount of chlorophyll in the wild-type cells, but a significant decrease in chlorophyll in the Sc∆CP12 cells (Figure 2). In addition, to compare the effect of high light stress on photosynthetic capacity, photosynthetic activity was compared. Since the high light treatment at 500 µmol photons m^−2^ s^−1^ had too strong an effect on the photosynthetic activity, the photosynthetic activity was compared using cells subjected to high light treatment at 300 µmol photons m^−2^ s^−1^. As a result, photosynthetic activity decreased only slightly even after 5 h of high light treatment in wild-type cells, but it decreased to 40% after 5 h of high light treatment in Sc∆CP12 cells (Figure 3). Comparing the amount of intracellular ROS at that time, it was clarified that ROS were significantly accumulated in the Sc∆CP12 cells as compared with the wild-type cells (Figure 4). It is widely known that under strong light conditions, reactive oxygen species are generated by the surplus photo-reducing power that is not used in the Calvin–Benson cycle and damage the photosynthetic system. However, 100 µmol photons m^−2^ s^−1^ of light is never too strong for cyanobacteria. Therefore, it is in a state where active oxygen is more likely to be generated inside Sc∆CP12 cells. Furthermore, when the effects of treatment with paraquat, which is an active oxygen generator, were compared between wild-type and Sc∆CP12 cells, significant chlorosis was observed in the Sc∆CP12 cells at the lower concentration of paraquat compared with wild-type cells (Figure 5). These data indicate that CP12 deficiency clearly reduces tolerance to photo-oxidative stress. In bacteria and archaea, there are various types of flavodiiron proteins (Flvs) that function as modular enzymes [24]. Some types of Flvs transfer the electrons of NADPH produced in Photosystem I (PSI) to oxygen and produce water without generating ROS, and so they are considered to function in oxidative stress protection in cyanobacteria [25,26]. CP12 may be involved in protection against oxidative stress by Flvs, or CP12 may suppress the generation of ROS by controlling the flow of electrons from PSI to NADPH because CP12 protein has the ability to bind to NADPH [9]. At this stage, there is no direct evidence to support this hypothesis, but it is worth further consideration given the emerging new function of CP12 in photosynthetic organisms.

## 4. Materials and Methods

### 4.1. Culture Conditions for Synechococcus Elongatus PCC7942

*S*. 7942 WT and CP12-deficient mutant (Sc∆CP12) [9] cells were cultured in 1 L of Allen’s medium at 27 °C under continuous illumination (40 or 100 µmol photons m^−2^ s^−1^) with the bubbling of sterile air at 8 L min^−1^.

### 4.2. Measurement of Photosynthesis and Chlorophyll Contents

Photosynthesis capacity was determined by measuring the rates of NaHCO_3_-dependent O_2_ evolution, as described previously [9]. Cultures in the log phase were washed and resuspended to an OD750 of 0.3 in fresh medium. An aliquot (1.0 mL) of these samples and 1 mM NaHCO_3_ were placed in a DW1 liquid-phase oxygen electrode chamber (Hansatech, Norfolk, UK) and stirred gently at 27 °C at 100 µmol m^−2^ s^−1^. The chlorophyll was extracted with acetone and determined content by spectrophotometer as described by Lichtenthaler [27].

### 4.3. Measurement of Intercellular ROS Level

Intercellular ROS levels were determined using the ROS Assay Kit—Highly Sensitive DCFH-DA—(Dojindo, Kumamoto, Japan). An aliquot (1.0 m) was collected and resuspended with Highly Sensitive DCFH-DA solution and incubated for 30 min under dark conditions, and then placed under 40 or 100 µmol photons m^−2^ s^−1^ conditions. Each cell was collected, washed, and resuspended with 100 µL of H_2_O, and then fluorescence was measured using GroMax Discover System (Promega, Tokyo, Japan) with excitation and emission wavelengths of 475 and 500–550 nm, respectively.

### 4.4. Data Analysis

Significance of differences between data sets was evaluated by *t*-test. Calculations were carried out with the Microsoft Excel software.

## Figures and Tables

**Figure 1 plants-10-01275-f001:**
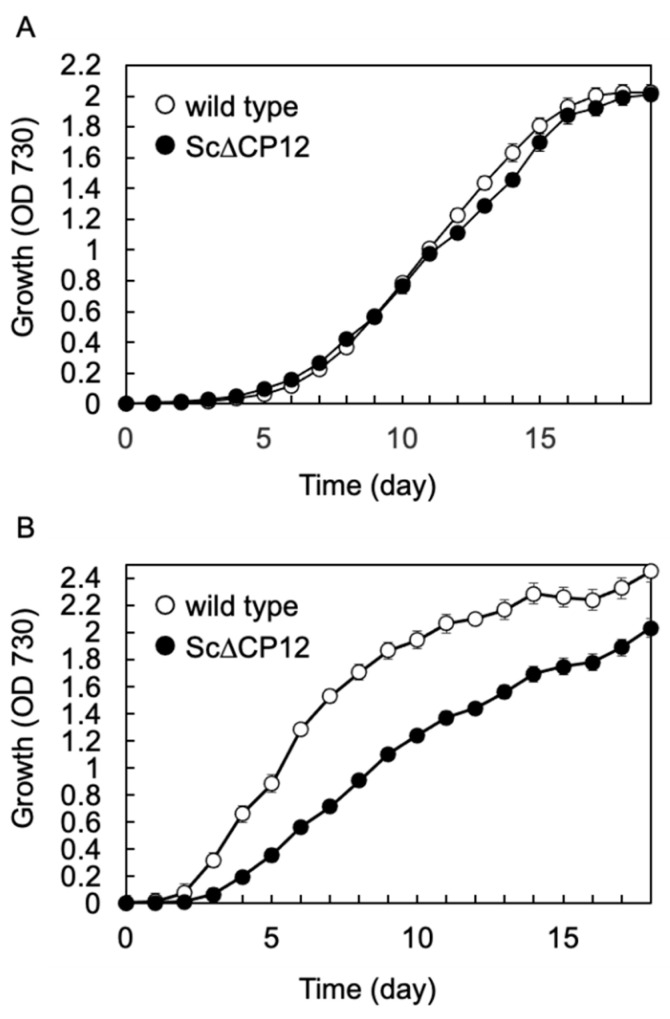
The time-course of photoautotrophic growth of wild-type and Sc∆CP12 mutant *S.* 7942 cells under continuous light conditions. Wild-type and Sc∆CP12 mutant *S.* 7942 cells were cultured in 1 L of Allen’s medium at 27 °C under continuous illumination (**A**): 40 µmol s^−1^ m^−2^, (**B**): 100 µmol s^−1^ m^−2^) with bubbling of sterile air at 1 L min^−1^.

**Figure 2 plants-10-01275-f002:**
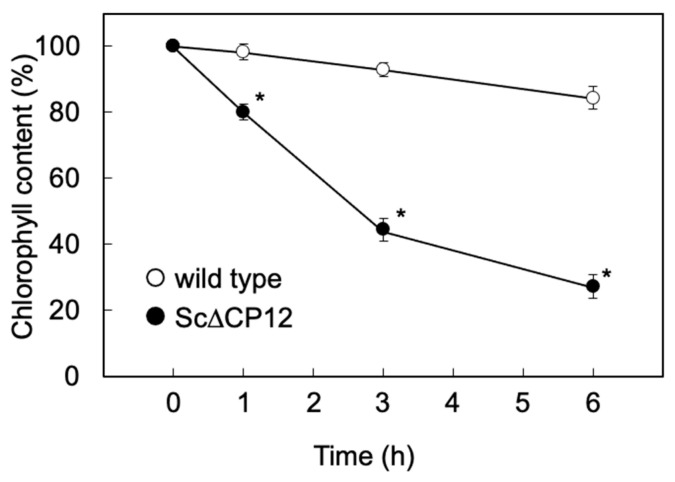
Effect of high-light treatment on the chlorophyll contents in wild-type and Sc∆CP12 mutant *S.* 7942 cells. Wild-type and Sc∆CP12 mutant *S.* 7942 cells were cultured in 1 L of Allen’s medium at 27 °C under continuous illumination (40 µmol s^−1^ m^−2^) with bubbling of sterile air at 1 L min^−1^ until the logarithmic growth phase, and then transferred to continuous illumination (500 µmol s^−1^ m^−2^). Values are as the mean ± standard error of four individual experiments. Asterisks indicate significant differences from the wild-type cells (* *p* < 0.01).

**Figure 3 plants-10-01275-f003:**
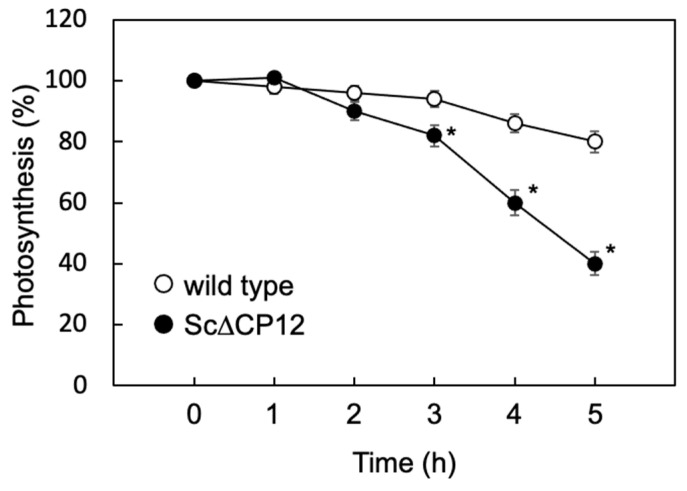
Effect of high-light treatment on the photosynthetic activity in wild-type and Sc∆CP12 mutant *S.* 7942 cells. Wild-type and Sc∆CP12 mutant *S.* 7942 cells were cultured in 1 L of Allen’s medium at 27 °C under continuous illumination (40 µmol s^−1^ m^−2^) with bubbling of sterile air at 1 L min^−1^ until the logarithmic growth phase, and then transferred to continuous illumination (200 µmol s^−1^ m^−2^). Changes in the oxygen evolution rate during incubation in the presence of 1 mM NaHCO_3_ under 100 µmol s^−1^ m^−2^ were measured using an oxygen electrode chamber, as described in Experimental Procedures. Values are the mean ± standard error of four individual experiments. Asterisks indicate significant differences from wild-type cells (* *p* < 0.01).

**Figure 4 plants-10-01275-f004:**
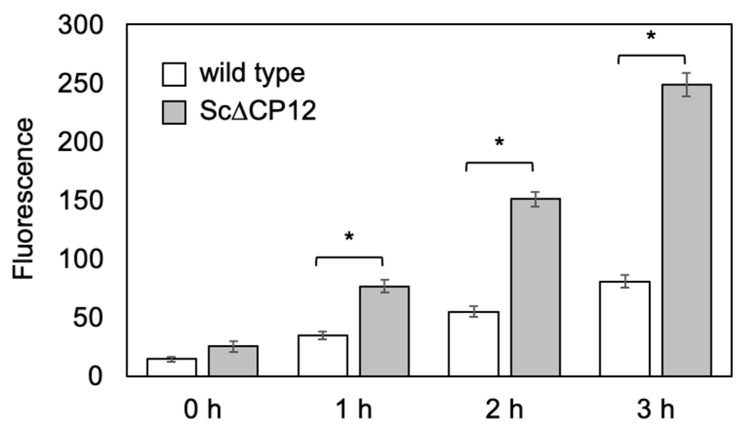
Intercellular ROS levels in wild-type and Sc∆CP12 mutant *S.* 7942 cells under high light conditions. Wild-type and Sc∆CP12 mutant *S.* 7942 cells were cultured in 1 L of Allen’s medium at 27 °C under continuous illumination (40 µmol s^−1^ m^−2^) with bubbling of sterile air at 1 L min^−1^ until the logarithmic growth phase, and then transferred to continuous illumination (200 µmol s^−1^ m^−2^). Intercellular ROS levels were measured as described in the Materials and Methods. Values are the mean ± standard error of four individual experiments. Asterisks indicate significant differences from the wild-type cells (* *p* < 0.01).

**Figure 5 plants-10-01275-f005:**
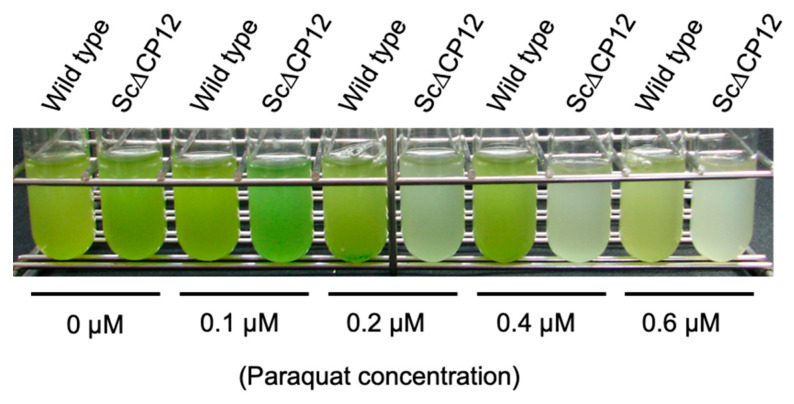
Effect of paraquat treatment on the growth of wild-type and Sc∆CP12 mutant *S.* 7942 cells. Wild-type and Sc∆CP12 mutant *S.* 7942 cells were cultured in 10 mL of Allen’s medium containing various concentrations of paraquat on a rotary shaker (120 rpm) under continuous light (40 μmol photons m^−2^ s^−1^) at 27 °C.

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
