# Peer review of "CP12 Is Involved in Protection against High Light Intensity by Suppressing the ROS Generation in Synechococcus elongatus PCC7942"

_plants, 2021, doi:10.3390/plants10071275_

Round 1

Reviewer 1 Report

CP12 is involved in protection against high light intensity by suppressing the ROS generation in Synechococcus elongatus PCC7942

CP12 is a small protein that forms a complex with GADPH and PRK in a NAD(P)H/NAD(P) dependent manner in darkness and thereby inactivates the CBB cycle in Synechococcus elongatus PCC7942. The authors characterized a CP12 deletion mutant and found that the strain was affected at high light intensities in comparison to the WT. Under high light the ScDCP12 strain was impaired in its growth, decreased its chlorophyll content and photosynthetic activity while intracellular ROS levels rose.

The authors convincingly show that CP12 has a potential function aside from GAPDH-CP12-PRK complex formation in darkness.

Minor suggestions:

Line 27/30 Calvin-Benson cycle/Calvin cycle; please stick either to Calvin-Benson or Calvin-Benson-Bassham cycle throughout the whole document.

Please include information from McFarlane et al. (2019): Structural basis of light induced redox regulation in the Calvin-Benson cycle in cyanobacteria, PNAS in the introduction

Line 56: it has been reported…

Line 65: cite Tamoi et al. 2005 for the ScDCP12 strain.

Lines 68 and 99: that ScDCP12 grows like the WT at 40 µmol light was already shown in Tamoi et al. 2005 (Fig. 6a). As you confirm this result here as a control, this confirmation should be mentioned and the article should be cited. The same applies to the results described in lines 99 and 100.

Line 70: …when grown at low….

Page 5; 2.5: It would be nice to explain in one sentence what paraquat does and with which purpose this experiment was performed.

Line 155 and 162: Please, be more precise here: ….affect photosynthetic activity in….under….conditions

Line 182: If I am not mistaken, it should be 200 µmol light instead of 300 µmol light. It is not clear to me here what is meant with “slightly lowered”. Lowered in comparison to what? Please clarify.

Author Response

Thank you very much for your treatment of the article that we submitted for publication in the Plants.  We have done our best to rewrite our article according to reviewer’s suggestions and comments.  We appreciate your kind and critical comments again.  We hope that these reversions are satisfactory and that the revised manuscript will be acceptable for publication in the Plants.

Minor suggestions:

Line 27/30 Calvin-Benson cycle/Calvin cycle; please stick either to Calvin-Benson or Calvin-Benson-Bassham cycle throughout the whole document.

RE: As Reviewer suggested, we have unified the term “Calvin-Benson cycle” throughout the document.

Please include information from McFarlane et al. (2019): Structural basis of light induced redox regulation in the Calvin-Benson cycle in cyanobacteria, PNAS in the introduction

RE: As Reviewer suggested, we have added a description of information from McFarlane et al. (2019) to the introduction at line 51-53.

Line 56: it has been reported…

RE: As Reviewer suggested, we have corrected.

Line 65: cite Tamoi et al. 2005 for the ScDCP12 strain.

Lines 68 and 99: that ScDCP12 grows like the WT at 40 μmol light was already shown in Tamoi et al. 2005 (Fig. 6a). As you confirm this result here as a control, this confirmation should be mentioned and the article should be cited. The same applies to the results described in lines 99 and 100.

RE: As Reviewer suggested, we have added the sentence and reference at line 67-71. 

Line 70: …when grown at low….

RE: As Reviewer suggested, we have corrected.

Page 5; 2.5: It would be nice to explain in one sentence what paraquat does and with which purpose this experiment was performed.

RE: As Reviewer suggested, we have added the sentence to explain paraquat at line 143-144.

Line 155 and 162: Please, be more precise here: ….affect photosynthetic activity in….under….conditions

RE: As Reviewer pointed out, we have revised the sentence to give a more polite explanation at line 161-163.

Line 182: If I am not mistaken, it should be 200 μmol light instead of 300 μmol light. It is not clear to me here what is meant with “slightly lowered”. Lowered in comparison to what? Please clarify.

RE: As Reviewer pointed out, we have revised the sentence to give a more polite explanation at line 189-191.

Reviewer 2 Report

The manuscript by Masahiro Tamoi and Shigeru Shigeoka concerns  the involvement of CP12 in the response to high light stress in cyanobacterium Synechococcus elongates. The Authors perfectly introduce in their research area, as well as the results are well discussed. The research objective is in the scope of Plants. However, I have some of the following minor comments:

The research hypothesis should be formulated - there is a weak justification for undertaking the research in the current version of MS

The description of Methods should be improved  - in addition to the reference, a "short description" could be included

I can not find a description of the statistical analysis used

The conclusion should be also improved.

In addition, due to the scope of the work performed, the paper should be assigned to a different type of article (short communication, etc.).

Minor:

Page 4, line 112 and 133 “were measured as described in the Experimental Procedures” – where is the section: “Experimental Procedures”?

Page 6, line 221 An aliquot (1.0m)?

Author Response

Thank you very much for your treatment of the article that we submitted for publication in the Plants.  We have done our best to rewrite our article according to reviewer’s suggestions and comments.  We appreciate your kind and critical comments again.  We hope that these reversions are satisfactory and that the revised manuscript will be acceptable for publication in the Plants.

The research hypothesis should be formulated - there is a weak justification for undertaking the research in the current version of MS

RE: As described in the introduction, CP12 is thought to function in controlling the activity of GAPDH and PRK, which function in the Calvin-Benson cycle, under dark conditions. However, in the analysis of CP12-deficient strains, it became clear that the growth of CP12-deficient strains was suppressed as the light became stronger. This suggests that CP12 works even under light irradiation. This study focuses on why CP12 deficiency causes growth inhibition under high light conditions.

The description of Methods should be improved – in addition to the reference, a "short description" could be included

RE: As Reviewer suggested, the explanation was added to the part where the explanation is insufficient at Materials and Methods.

I can not find a description of the statistical analysis used

RE: As Reviewer suggested, we have added the description of the statistical analysis used at line 239-241.

The conclusion should be also improved.

RE: The Instructions for Author stated that conclusion is non-essential, so I didn't write it in this manuscript.

In addition, due to the scope of the work performed, the paper should be assigned to a different type of article (short communication, etc.).

RE: The editor commented that "communication" was suitable for this treatise. However, if the editor determines that "Short communication" is more suitable, we will change it.

Minor:

Page 4, line 112 and 133 “were measured as described in the Experimental Procedures” – where is the section: “Experimental Procedures”?

RE: We corrected to “Materials and Methods”.

Page 6, line 221 An aliquot (1.0m)?

RE: We are sorry mistyped.  We corrected to 1 mL.

Reviewer 3 Report

The Authors presented us a professional and interesting plant/cell physiological work. The MS is of very high quality, at the level of an already published paper. The Authors found a surprising phenomenon and resolved it using well-conceived research plan and appropriate methods which led to novel and interesting results. I congratulate the Authors and wish them further success in future works.

Author Response

Thank you very much for your treatment of the article that we submitted for publication in the Plants.  We have done our best to rewrite our article according to other reviewer’s suggestions and comments.  We appreciate your kind and critical comments again.  We hope that these reversions are satisfactory and that the revised manuscript will be acceptable for publication in the Plants.